# Coherent microwave comb generation via the Josephson effect

Angelo Greco ✉, Xavier Ballu, Francesco Giazotto ⬡ & Alessandro Crippa ⬡ ✉

Frequency combs represent exceptionally precise measurement tools due to the coherence of their spectral lines. While optical frequency comb sources constitute a well-established technology, superconducting circuits provide a relatively unexplored on-chip platform for low-dissipation comb emitters able to span from gigahertz to terahertz frequencies. We demonstrate coherent microwave frequency comb generation by leveraging the ac Josephson effect in a superconducting quantum interference device. A time-dependent magnetic drive periodically generates voltage pulses, which in the frequency domain correspond to a comb with dozens of spectral modes here reported up to mode 46. The emitted power at the device level ranges from −170 dBm to −130 dBm per harmonic, corresponding to 40 dB dynamic range in the 4-8 GHz bandwidth. The micrometer-scale footprint and minimal dissipation inherent to superconducting systems foster the integration of our comb generator with advanced cryogenic electronics. Transferring optical techniques to the solid-state domain may enable new applications in quantum technologies.

Time-frequency duality implies that a periodic signal in time corresponds to one or more equally spaced spectral lines in the frequency domain. This spectrum is referred to as a frequency comb when a stationary phase relation is established among the frequency lines. A variety of methods to generate frequency combs[1,2] have found applications in communications, spectroscopy, frequency metrology, optical clocks, computing, and quantum information, spanning from near-infrared to ultraviolet regions of the electromagnetic spectrum[3]. In condensed matter, researchers have utilized mesoscopic devices governed by quantum phenomena as multitone synthesizers[4,5] and coherent photon emitters[6–8], revealing lasing properties[9–11] and leading to various implementations in quantum optics at a microscopic scale[12]. However, a tunable cryogenic source of broadband microwave combs in the frequency range where commercial electronics operate is still lacking. As many superconducting and spin-based quantum bits (qubits) need microwave radiation below 8 GHz for coherent control and dispersive readout[13,14], compact on-chip frequency comb generators could act as cryogenic frequency up-converters to significantly decrease the number of interconnects between quantum processors and room temperature electronics.

To date, the few examples of cryogenic combs in the microwave range rely upon long chains of Josephson junctions[15] or are embedded in cavities[16–20], which have a bandwidth that is hardly tunable in situ and constrict the repetition frequency[21]. Moreover, the millimeter-size footprint of such devices presents significant limitations to scalability and integration with current cryogenic electronics.

Here, we demonstrate the generation of frequency combs throughout the entire C-band frequency domain (4-8 GHz) by magnetically driving a dc Superconducting Quantum Interference Device (SQUID). A time-dependent magnetic flux with frequency $f_p$ modulates the gauge-invariant phase across the SQUID, which leads to sharp voltage pulses generated at a fixed repetition rate. This train of short pulses ultimately results in a frequency comb spectrum with $n$ modes at frequencies $2f_p$, $3f_p$, ..., $nf_p$, where $n > 50$.

We emphasize that pulse synthesis here does not rely on cavities, unlike conventional optical frequency combs. This leads to few key differences. In cavity-based systems, cavity length and group velocity set the carrier frequency and the pulses repetition rate, whereas in our case the pump frequency $f_p$ is freely tunable, potentially allowing harmonic generation from gigahertz to terahertz frequencies. Moreover, while cavity dispersion induces a finite carrier-envelope frequency offset in the spectrum, our architecture yields an offset frequency of zero, since there is no cavity and, therefore, no carrier frequency inside the voltage pulses.

NEST, Istituto Nanoscienze-CNR and Scuola Normale Superiore, Pisa, Italy. ✉e-mail: angelo.greco@nano.cnr.it; alessandro.crippa@cnr.it

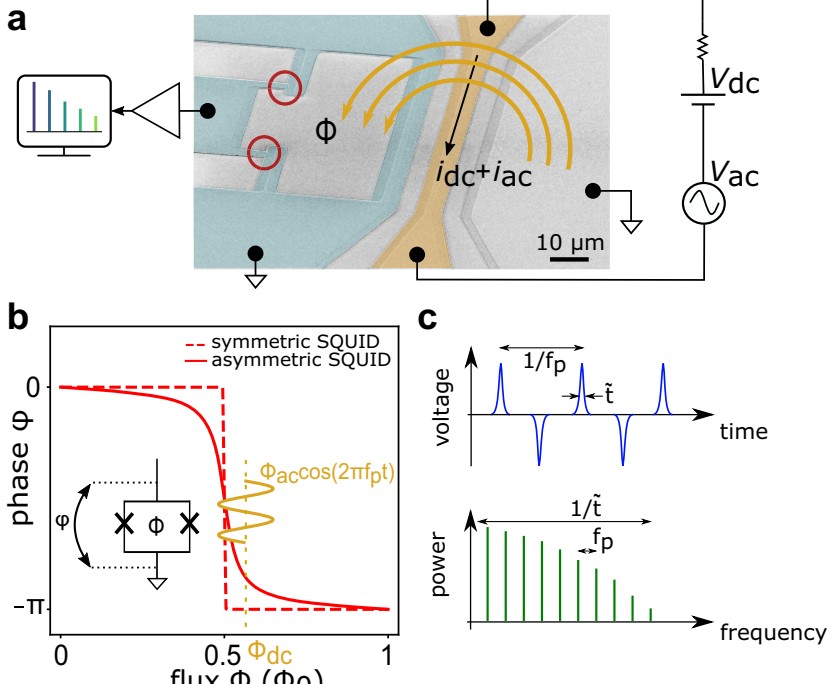

**Fig. 1 | SQUID-based microwave comb generator. a** Scanning electron micrograph of the device with false colors. A time-dependent current with a static bias flows through the coplanar waveguide (in yellow), generating a magnetic flux $\Phi$ threading the dc SQUID (in cyan) loop. The red circles indicate the locations of the two Josephson junctions. The train of voltage pulses produced across the SQUID is transmitted by a 50 $\Omega$ transmission line to the amplification chain. **b** Phase-flux dependency underlying the device's operating principle for a symmetric SQUID (dashed trace, $r \simeq 0$) and a strongly asymmetric SQUID (solid line, $r = 0.1$). Around $\Phi_0/2$, the ac flux with frequency $f_p$ sweeps the phase $\varphi$ across the SQUID, as illustrated in the inset. **c** Upper panel: Time-domain representation of a train of voltage pulses generated by the SQUID with a repetition time $1/f_p$. Each pulse has a width of $\tilde{t}$. Lower panel: Sketch of the frequency comb spectrum corresponding to the above voltage signal. The mode spacing is $f_p$, i.e., the inverse of the repetition time, and the width of the spectral envelope, $1/\tilde{t}$, is on the order of the inverse of the pulse duration.

The comb source consists of a dc SQUID made of two aluminum Josephson tunnel junctions connected in parallel to form a loop. One side of the SQUID is grounded, while the other side is linked to a coplanar waveguide (CPW) to transmit the voltage pulses to the amplification chain (Fig. 1a). The signal is further amplified at room temperature and acquired by either a spectrum analyzer or a high-frequency lock-in amplifier. Supplementary Note 4 reports a complete description of the measurement setup. A second CPW, inductively coupled to the loop, provides the magnetic flux threading the SQUID.

We present measurements on two samples (see "Methods" and Supplementary Note 6 for fabrication details) with nominally identical loop geometry and Josephson junction characteristics but differing in the coupling strength with the drive tone. Sample 1 has a flux line 9 µm wide with a flux line−loop mutual inductance of 4.5 pH, while Sample 2 has a flux line 3 µm wide and a mutual inductance of 0.44 pH. Sample 1 spans multiple periods of the SQUID flux characteristic without turning the flux line normal, while Sample 2 is conceived for high-resolution measurements (in Methods the parameters of both devices). Unless otherwise specified, we report data from Sample 1, collected at a temperature of 60 mK in a dilution refrigerator.

To illustrate the working principle of our device[22–24], we first consider the general equation of the Josephson current through a dc SQUID (see Supplementary Note 7):

$$I_J = I_+ (\cos \phi \sin \varphi + r \sin \phi \cos \varphi), \tag{1}$$

with $I_+ = I_{c1} + I_{c2}$ representing the sum of the critical currents of the two junctions, $\varphi = (\varphi_1 + \varphi_2)/2$ is the average phase drop across the interferometer, $\phi = \pi \Phi/\Phi_0$ with $\Phi$ the magnetic flux through the loop and $\Phi_0 \simeq 2 \times 10^{-15}$ Wb the flux quantum, and $r = (I_{c1} - I_{c2})/(I_{c1} + I_{c2})$ indicates the degree of asymmetry of the junctions. Screening effects caused by significant loop inductance have been neglected.

In our setup, since no dc bias current is imposed across the SQUID ($I_J = 0$), Eq. (1) can be rewritten as $\varphi = \arctan(-r \tan \phi)$. Figure 1b displays the evolution of this phase-flux relation over a period $\Phi_0$. At $\Phi_0/2$, the number of flux quanta enclosed by the superconducting loop changes by one, and the energy is minimized by adjusting the phase $\varphi$ across the SQUID in a way that depends on the symmetry parameter $r$. In the case of identical junctions, where $r \simeq 0$, $\varphi$ varies abruptly when the external flux $\Phi$ is swept across $\Phi_0/2$. For SQUIDs with asymmetric branches, as in our experiment, $r \neq 0$ and the dependency gets smoothed near $\Phi_0/2$, reducing the phase modulation speed.

## Results

To operate the device as a comb source, we apply a time-dependent flux $\Phi_{ac} \cos(2\pi f_p t)$ superimposed on a static flux bias $\Phi_{dc}$ using the inductively coupled CPW line (Fig. 1a). For $\Phi_{dc} \approx \Phi_0/2$, the oscillating flux periodically sweeps the phase $\varphi$ (Fig. 1b). Due to the ac Josephson effect, the time modulation of $\varphi$ establishes a finite voltage $V$ across the SQUID. This leads to a sequence of voltage pulses with alternating signs[22], as illustrated in the upper panel of Fig. 1c and confirmed by simulations (Supplementary Note 3). This signal corresponds to a series of spectral lines in the frequency domain with a frequency spacing determined by the pulse repetition rate, as shown in the lower panel of Fig. 1c. The steepness of the flux-phase relation sets the duration of each voltage pulse, which ultimately establishes the harmonic content of the comb.

In optics, a frequency comb is a spectrum of phase-coherent, evenly spaced narrow lines. In the following, we demonstrate that the signal generated by our device meets the criteria of a frequency comb:

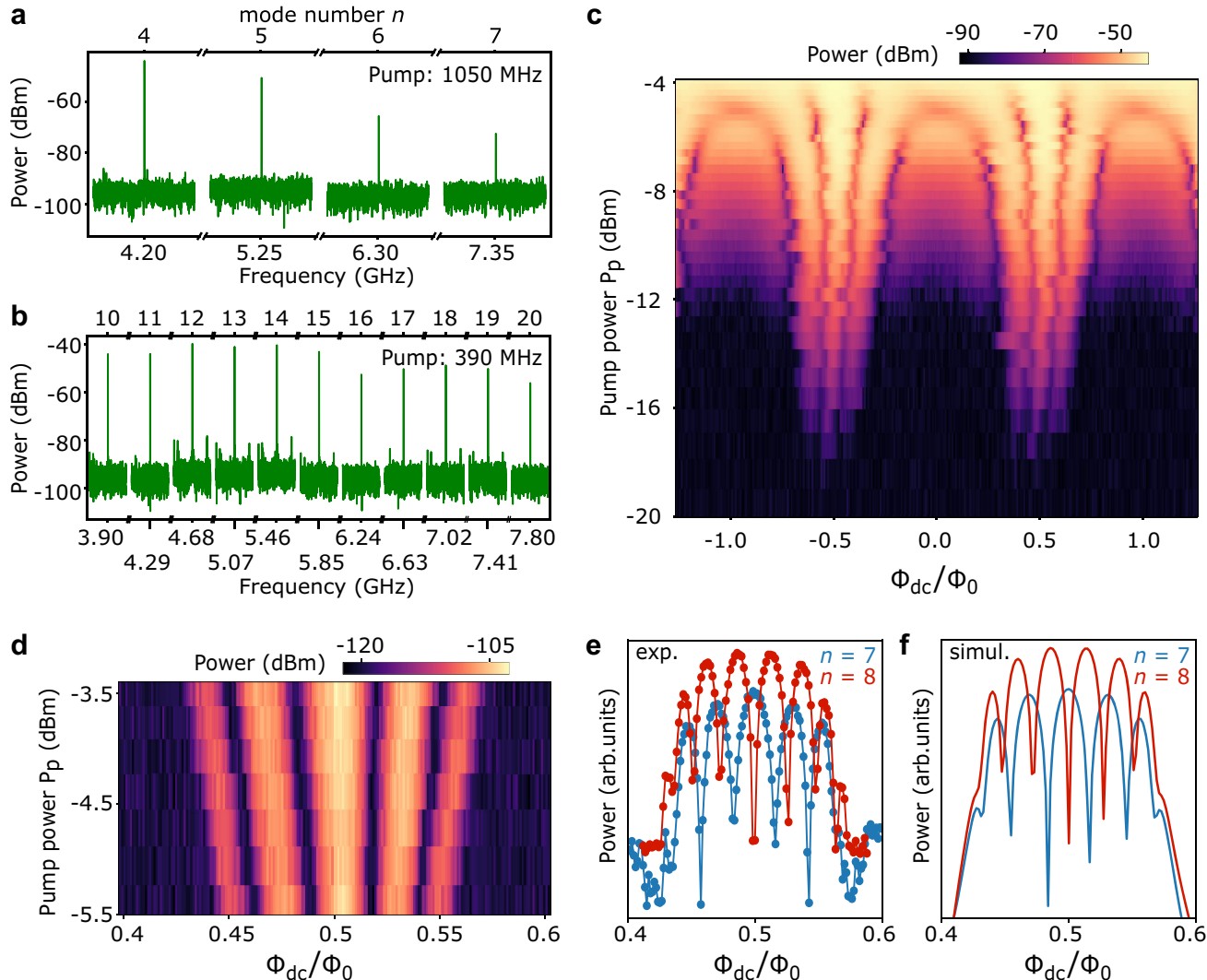

**Fig. 2 | Frequency comb spectrum. a** Power spectrum from various spectra, each sampled with a 20 kHz span. The spikes are evenly spaced and correspond to multiples from $n = 4$ to $n = 7$ of the pump frequency $f_p = 1050$ MHz and pump power −8 dBm. **b** Same as (**a**), with harmonics from 10 to 20 of $f_p = 390$ MHz and pump power 0 dBm. **c** Output power of the 5th harmonic with $f_p = 833.34$ MHz as a function of flux bias $\Phi_{dc}$ and nominal power of the pump tone $P_p$. Each data point reports the maximum of a spectral trace spanning 500 Hz. The 5th mode

(4166.7 MHz) lies in the setup bandwidth with highest gain. **d** Zoom-in of the output power of the 7th harmonic generated by Sample 2. The pump frequency is 597.34 MHz, and the resulting signal is at 4181.38 MHz. **e** Output power of the 7th (blue) and 8th (red) harmonics generated by Sample 2 under the same conditions as (**d**) at fixed pump power. **f** SPICE simulation of output power for the 7th and 8th harmonics relative to (**e**).

first, we address the equidistance of the lines; next, we evaluate the coherence of individual modes; and finally, we demonstrate phase control of the modes and measure their mutual phase relation.

Later on, we shall refer to the harmonics power as the power recorded at the acquisition step, i.e., the power emitted by the device plus an instrumental gain which varies approximately linearly from ~87 dB at 4 GHz to ~82 dB at 8 GHz (Supplementary Note 4 provides the complete setup).

Figure 2a, b display a segment of the comb spectrum within the C-band for pump frequencies of 1.050 GHz and 390 MHz, respectively. Both signals exhibit sharp frequency lines corresponding to integer multiples of the pump frequency,

$$f_n = n \times f_p. \tag{2}$$

The power amplitude associated with the $n$-th harmonic generally decreases with increasing $n$ due to higher attenuation in the readout circuitry and reduced up-conversion efficiency[22].

To characterize the flux response of our comb source, Fig. 2c reports the power of the 5th harmonic of a pump tone 833.34 MHz as a function of the nominal pump power $P_p$ (i.e., prior to attenuation, see Supplementary Note 4) and the magnetic bias $\Phi_{dc}$, which tune the ac amplitude and the offset of the flux, respectively. As expected, the intensity of the harmonic displays an overall periodic pattern, a hallmark of flux-modulated SQUIDs. The flux period aligns with that of the reflectometry trace of the same device (Supplementary Note 1 and Supplementary Fig. 1). As the pump power increases, the flux wiggles cover a larger portion of the phase-flux relation, broadening the flux range suitable to signal generation. A sizable signal appears in between the lobes for $P_p > -12$ dBm, probably due to finite phase rolling over the entire flux period for asymmetric SQUIDs (Fig. 1b).

When $P_p \geq -5$ dBm, $\Phi_{ac}$ reaches and exceeds $\Phi_0$. In this regime, a finite signal is generated for any value of $\Phi_{dc}$, yielding more than two pulses per period of the drive. Overall, Fig. 2c displays a dynamic range of 40 dB of the output power, which at the device level is estimated to span from −170 dBm to −130 dBm.

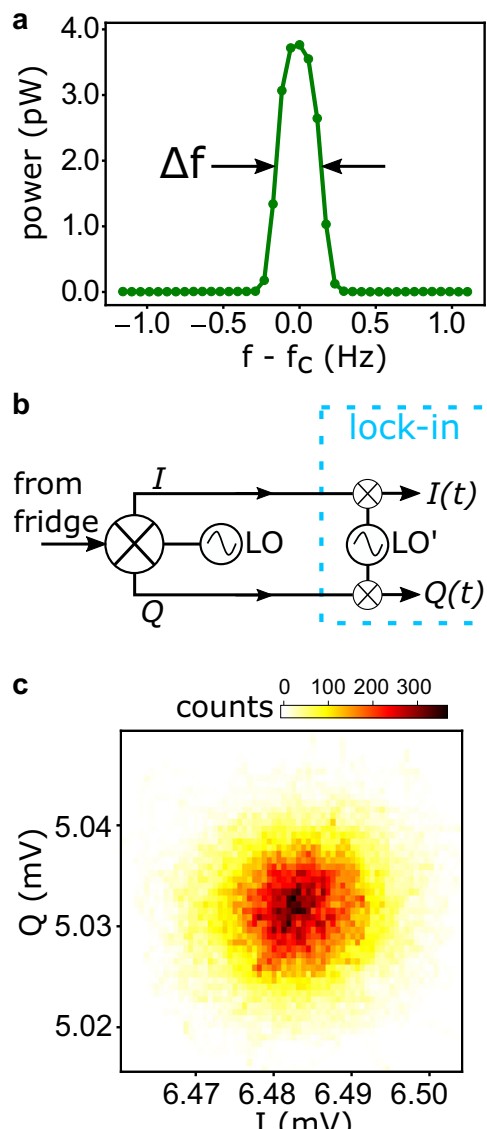

**Fig. 3 | Single-mode linewidth and time stability. a** Power spectrum of the 5th harmonic with $f_c = 5 \times f_p$ ($f_p = 833.34$ MHz) measured at $\Phi_0/2$. The full-width at half-maximum of the spectral line profile $\Delta f$ coincides with the instrumental resolution bandwidth (0.3 Hz), establishing a lower bound. **b** Schematic diagram of the heterodyne setup to acquire the two quadratures in the time domain, $I(t)$ and $Q(t)$, of the comb mode in (**a**). **c** Histogram of the two quadratures over a 10-s acquisition time.

Figure 2d provides a close-up view around $\Phi_0/2$ of Sample 2 for $\Phi_{ac} \ll \Phi_0$. We observe distinct interference minima, whose number and flux offset position depend on the harmonic index and parity. In the line cuts of Fig. 2e, f, experimental measurements demonstrate excellent agreement with numerical simulations performed using SPICE-based circuit models described in Methods. Lastly, the harmonic generation is very robust in temperature, as appears in Supplementary Fig. 2.

Having clarified the impact of flux bias and pump power on the generated signal, we now assess the coherence of a single mode of the comb. Figure 3a presents a high-resolution spectrum of the 5th harmonic of a comb obtained by using a pump tone of $f_p = 833.34$ MHz, at a power of −17 dBm (meaning $\Phi_{ac} < \Phi_0$, as in Fig. 2b) with $\Phi_{dc} \simeq \Phi_0/2$. The peak is centered at $5 \times f_p$ and shows a full width at half maximum $\Delta f \simeq 0.34$ Hz, which corresponds to a coherence time $1/\Delta f \simeq 3$ s. We stress that this value is as a lower limit for the actual coherence time,

since the linewidth of the comb mode cannot be resolved with our spectrum analyzer, having a minimum resolution bandwidth of 0.3 Hz.

To further confirm that the single-mode coherence lasts for seconds, we probe the harmonic quadratures over time. First, we downconvert the comb signal using a heterodyne setup, then we demodulate it with a high-frequency lock-in, which digitizes the in-phase $I(t)$ and quadrature $Q(t)$ components over a 10-s time window (see Fig. 3b). Figure 3c shows the two-dimensional histogram of the $I$ and $Q$ samples. The standard deviation of the counts measures $\simeq 20\,\mu$V along both quadratures, which certifies the time stability of the mode phase with respect to the local oscillators of the circuitry (Fig. 3b). In our setup, the broadening of such kind of distributions is dominated by the noise of the readout chain, see Supplementary Fig. 6.

We have demonstrated that the spectral modes of our frequency comb follow the simple relation of Eq. (2). It can be shown (see "Methods") that in general a similar relation holds for the phase $\theta_n = n \times \theta_p + \widetilde{\theta}_n$, where $\theta_n$ is the phase of the $n$-th mode, $\theta_p$ is the phase of the pump tone, and $\widetilde{\theta}_n$ is an offset of the specific mode. This means that the phase of a mode varies according to the mode order when the pump phase is changed. We prove it by means of the setup shown in Fig. 4a, where a phase shifter to the pump tone $\theta_p$ is added. The generated comb signal is then acquired in both quadratures, similarly to Fig. 3b. Figure 4b reports the IQ histograms of the 9th harmonic with $f_p = 597$ MHz for eight values of $\theta_p$, shifted by 5 deg at each step. The eight resulting point clouds are rotated by 45 deg each other, implying that $\delta\theta_9 = 9 \times \delta\theta_p$ ($\delta$ denotes the phase difference).

This property holds for any harmonics of the comb spectrum. As examples, we choose a pump tone at $f_p = 121$ MHz, and we address the phase of the modes from 42 to 46 by the frequency-multiplexed demodulation setup in the schematic of Fig. 4c.

The experimental outcome is shown in Figure 4d, in terms of phase of such five harmonics as a function of the pump phase $\theta_p$. The phase of each mode evolves linearly with respect to pump phase (to make it more visible, the phase data have been rescaled differently for each mode, while the intercepts $\widetilde{\theta}$ have been shifted to zero).

A further consistency check is to verify that

$$\theta_{n+1} - \theta_n = \theta_p + \widetilde{\theta}_{n+1} - \widetilde{\theta}_n, \tag{3}$$

meaning that the phase difference between two subsequent modes is always equal to the pump phase $\theta_p$ plus an offset. The inset of Fig. 4d shows $(\theta_{43} - \theta_{42})$ vs $\theta_p$ (offset removed for clarity). The linear fit yields a slope of 1, disclosing the constraint between the phases of the harmonics (Eq. (3)), which, together with the equidistance of frequencies (Eq. (2)), is peculiar of a comb spectrum.

## Discussion

Finally, some considerations about the dissipated power and maximum frequencies of operation are worth discussing.

We estimate the energy dissipation due to Joule heating to be $\sim 2 \times 10^{-26}$ J per pulse (see "Methods"), which yields a power dissipation of $10^{-18}$ W at 100 MHz pump frequency. Such a value is extremely low because the device never switches to the normal state, in stark contrast with some other superconducting technologies[25,26].

The micrometer-size and the number of generated modes make our platform suitable for scalability, while the minimal dissipated power allows on-chip integration with other quantum systems sensitive to quasi-particles generation, such as qubits or sensors. In the present experiment, the power dissipation at the coldest plate is primarily due to the flux drive (1 μW at most, see "Methods"), which can be reduced by enhancing the inductive coupling between flux line and SQUID loop, while the contribution of the device is negligible. Such a heat load is largely sustainable by current dilution refrigerators with typical cooling power ranging from from 0.1 mW to few mW[27]. This

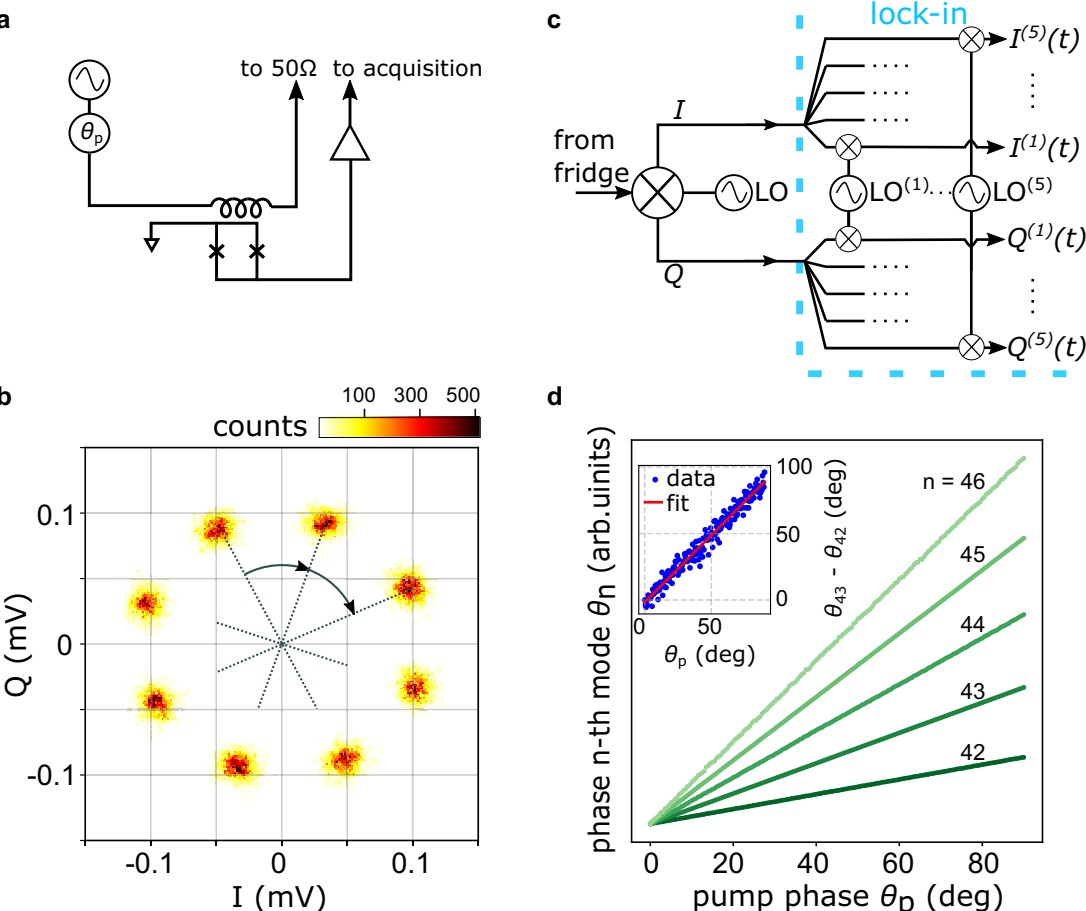

**Fig. 4 | Phase tunability and mutual phase of comb modes. a** Measurement setup for tuning the phase of comb modes by the pump phase $\theta_p$. The phase of the comb modes is addressed by the two-step down-conversion circuit of Fig. 3b. **b** Histograms of 16 s-long measurements of the 9th harmonic generated by $f_p$ = 597 MHz with power $P_p$ = −5 dBm. The data are clustered into eight point clouds, corresponding to the values of $\theta_p$ stepped by 5 deg every 2 s. Each variation of $\theta_p$ by 5 deg leads to a 45 deg tilt in the IQ plane. **c** Demodulation circuit to acquire both quadratures of five comb modes referenced to the same clock and sampled simultaneously. **d** Phases of comb modes from 42 to 46 as a function of the pump phase acquired with the demodulation circuit of (**c**). Each dataset is rescaled by a multiplication factor for clarity and the intercepts of the curves are set to zero to highlight the relative slopes. Inset: linear fit of experimental points obtained by evaluating difference between the phase of mode 43 ($\theta_{43}$) and the phase of mode 42 ($\theta_{42}$). The fit yields a slope of 1.00 ± 0.01 and an intercept of −1.56 ± 0.71.

offers a significant advantage over cryogenic Complementary Metal Oxide Semiconductor (cryo-CMOS) circuits, whose dissipation is of the order of tens of mW for 1 GHz clock frequencies[28].

There are then some limitations concerning both the maximum harmonic frequency and pump frequency usable for the ac flux.

The former can extend up to the depairing frequency, i.e., the photon frequency at which the Cooper pairs are broken. It is directly related to the superconducting gap, and as an example it is about 50 GHz for aluminum and higher than 300 GHz for niobium.

The highest pump frequency is mainly related to the plasma frequency of the SQUID. This bottleneck concerns the capability of the superconducting phase to follow rapid flux variations, up to timescales limited by the plasma frequency (usually between 10 GHz and 100 GHz).

Our results demonstrate the operation of a dc SQUID with a time-dependent magnetic drive as a frequency comb source in the microwave C-band. Contrary to previous demonstrations, the comb spectrum is generated without any resonator to be seeded to accomplish four-wave mixing[16] or whose emission has to be stabilized via injection-locking[10,19]. The tunability of the comb emitted opens up the possibility of exploiting optical techniques, e. g. frequency comb spectroscopy[3], to selectively address integrated sensors, and perform multiqubit entangling operations[29] or readout.

In the future, we aim to boost the output power by using dc SQUIDs with tunable symmetry (recall Fig. 1b), for instance via electrostatic gating of semiconducting weak links, or by means of other geometries as rf SQUIDs, where the phase-flux relation around half flux quantum can be very steep next to the crossover to the hysteretic regime, and finally via optimized flux signals.

In conclusion, features as μm-size physical footprint, low latency, and significant operational bandwidth make the Josephson radiation comb generator suitable for integration into classical interfaces for quantum sensing and computing.

## Methods
### Devices fabrication
The lithographic mask for our frequency comb was realized using a single step of optical lithography on a double-layer photoresist substrate composed of S1805 on top of LOR20B. The Al deposition was performed in an UHV electron-beam evaporator through shadow-mask evaporation using two different angles. In the first evaporation, we deposited the counter electrode of 30 nm at a tilting angle of 20°, then rotating the sample holder to −20° angle, we evaporated the second layer of 100 nm to realize the top electrode. The growth of the barrier in between the two evaporations was obtained by thermal oxidation at room temperature of the counter electrode in the

loadlock of the deposition system. The oxidation lasted for 40 min at 15 Torr of pure oxygen.

By construction in both devices, the self-inductance of the SQUID loop is 100 pH, while the critical currents range between 100 nA and 200 nA (range given by the fabrication dispersion of tunnel junctions parameters.) Sample 1 has a mutual inductance between the flux line and the loop of 4.5 pH, while Sample 2 of 0.44 pH. The difference in mutual inductance is obtained by modifying the width of the flux line next to the SQUID loop. Supplementary Fig. 7 and Supplementary Fig. 8 report the mask layout and a SEM picture enclosing the active area of the device, respectively.

### Frequency comb equation and phases of comb modes

The electric field $E$ of a frequency comb can be expressed by the equation[30]

$$E(t) = \sum_n E_n e^{i[2\pi(f_n + f_{ceo})t + \widetilde{\theta}_n]}, \tag{4}$$

where $E_n$ is the amplitude of the $n$-th mode, $f_n$ is its frequency, $\widetilde{\theta}_n$ its phase offset and $f_{ceo}$ is the carrier-envelope offset frequency. If all the modes lie at integer multiples of the fundamental frequency, as in our case, $f_{ceo} = 0$, hence the frequency of the $n$-th mode can be simply written as $f_n = n \times f_p$. As such, the phase of a generic mode can be expressed as $\theta_n = n \times \theta_p + \widetilde{\theta}_n$, which implies that the phase of the $n$-th mode can be tuned by the phase of the pump $\theta_p$. As a consequence, the difference in phase between any two subsequent modes is $\theta_{n+1} - \theta_n = \theta_p + \widetilde{\theta}_{n+1} - \widetilde{\theta}_n$.

### Dissipated power

In the following, we estimate the dissipated power during operation by Joule heating. Each pulse generated by the SQUID imposes a voltage drop across it, meaning that a portion of the energy coming from the pump is dissipated into heat. The fraction of dissipated energy can be estimated by considering the RCSJ model of the SQUID. In this model the shunting resistance is given by the differential resistance of the Josephson tunnel junctions. This load can be approximated by considering it equal to the subgap resistance $R_{SG}$ for an average voltage drop lower than the critical voltage $V_c = I_c R_N$, and equal to $R_N$ for voltage drops higher than $V_c$. In the perfectly symmetric case, our device generates theoretical voltage spikes with maximum height in the order of $V_c$. This value drops in the physical case of finite asymmetry, letting us saying that the average voltage drop for each spike is below $V_c$. We can hence estimate the dissipated energy in our case (Sample 1, circuit parameters from reflectometry fit and room temperature measurement) per pulse by considering a voltage spike of average height 2 μV and temporal width $\Delta t \approx 500$ ps according to simulations of Supplementary Note 3. The average power dissipated on a single junction during a spiking event is $P \approx V^2/R_{SG} \approx 2 \times 10^{-17}$ W, where $R_{SG}$ is typically $\gtrsim 10^2 R_N$ in Al S-I-S junctions[31] and we have considered the lower bound $R_{SG} = 10^2 R_N \approx 200$ kΩ. Therefore, the dissipated energy per pulse is $P \times \Delta t \approx 1 \cdot 10^{-26}$ J. We can finally estimate the dissipated energy per pulse times the pump frequency (100 MHz in this case). By summing the power dissipated from both junctions, this yields $P_{heat} \approx 2 \times 10^{-18}$ W.

The dissipation due to Joule heating by the two Josephson junctions per pulse is a figure of merit of primary relevance to interface on-chip the comb source with other quantum systems. However, additional sources of dissipation are present when considering the setup beyond the SQUID itself, such as the static power dissipation associated to the current across the resistive network in the drive line circuitry. For instance, the dc flux bias is provided to the sample by a coaxial line with LC and IR filters interposed at the mixing chamber stage, see Supplementary Fig. 5. The ohmic resistance of the flux line is 40 Ω from room temperature to the mixing chamber plate, including semi-attenuated coax

cables, low-pass filters and IR filters. A typical current bias of 100 μA yields a dissipated power of 0.4 μW across the whole cryostat. About the ac component of the flux, the coaxial cables from room temperature down to the coldest plate attenuate 14–20 dB within the 0.2–1 GHz bandwidth. At the mixing chamber stage, after low-pass and infrared (IR) filtering, copper coaxial lines connect the chip carrier with negligible attenuation. Altogether, in this case the dissipation is of the order of tens of μW throughout the fridge, with much less than 1 μW at the mixing chamber plate (a 0.1 dB dissipation by a IR filter of a −20 dBm signal yields a dissipation of 0.23 μW).

It is worth mentioning that a way to reduce the current through the drive line, and therefore the overall dissipation, is to enhance the magnetic coupling of the flux line to the SQUID loop, for instance by on-chip spiral inductors patterned on top of the SQUID.

## Data availability

The raw data that support the findings of this study are available from the corresponding authors upon request. https://doi.org/10.5281/zenodo.18403390.

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

## Acknowledgements

We thank Max Hofheinz, Paolo Solinas, Leonardo Viti, Riya Sett, and Alessandro Tredicucci for helpful discussions and availability. We also acknowledge Fabio Tramonti of the electronic workshop at INFN Pisa for assistance. A.G., F.G. and A.C. acknowledge support by the EU's Horizon 2020 Research and Innovation Framework Program under Grant Agreement No. 964398 (SUPERGATE) and No. 101057977 (SPECTRUM). A.G. and F.G. acknowledge the PNRR MUR project PE0000023-NQSTI. X.B. and A.C. acknowledge support by the European Union NextGenerationEU Mission 4 Component 1 CUP B53D23004030006 PRIN project 2022A8CJP3 (GAMESQUAD).

## Author contributions

A.G., X.B., and A.C. performed the measurements. A.G. and A.C. conceived and designed the experiment from an idea of F.G. A.G. fabricated the samples. A.G. and A.C. analyzed the results and wrote the manuscript with inputs from all authors.

## Competing interests

A.G., F.G. and A.C. have filed a provisional patent application that relates to the Josephson comb generator. X.B. declares no competing interests.
