## [Transparent Peer Review file · Nature Communications]

Coherent microwave comb generation via the Josephson effect

Corresponding Author: Dr Alessandro Crippa

Version 1:

Reviewer comments:

Reviewer #1

(Remarks to the Author)

The manuscript by Angelo Greco et al. presents a novel approach to generating a microwave frequency comb based on superconducting circuits. The authors demonstrate that frequency combs with well-detuned comb spacing and a broad frequency range can be achieved by periodically modulating the flux across a SQUID. They characterize the comb emission by varying the magnetic flux modulation intensity and bias settings. The combs exhibit narrow linewidths and good coherence for the single mode, as well as well-defined mutual coherence among different modes, suggesting potential applications across a range of fields.

This work is grounded in the AC Josephson effect in SQUIDs, a well-established phenomenon that has been extensively reported in the literature. The authors employ a periodic signal to induce phase slips, which generate voltage pulses that naturally form a frequency comb. This outcome, while technically sound, is not unexpected. A significant portion of the manuscript focuses on characterizing the frequency comb in an attempt to demonstrate its excellent coherence. In principle, a train of ideal voltage pulses would yield an ideal frequency comb, which is also a well-known result. As such, the coherence of the experimentally synthesized frequency comb is primarily determined by the coherence of the flux drive signal and the stability of the DC bias. With this in mind, I have concerns regarding the characterization of the comb's coherence, as detailed in the following questions.

While the topic—generating microwave frequency combs with chip-scale devices—is certainly of broad interest, I am not convinced that the work presents a level of novelty that justifies publication in Nature Communications.

Additionally, I have several questions and points for further clarification:

1. Normalization of Phi Values: In the manuscript, the authors normalize ϕ_{DC} to ϕ_0 , but use units of dBm for ϕ_{AC} , which may cause confusion. Would it be possible to calibrate ϕ_{AC} also to ϕ_0 in order to provide a more consistent description?
2. is it $\phi_{AC} \gg \phi_0$?
3. Line 176: It would be helpful to list the value of ϕ_{AC} here as well, for completeness.
4. From line 189 to line 193 & Fig. 3, the authors claim the coherence of a single mode of the frequency comb based on the shape of the I-Q histogram, which is one of the key results. However, this discussion raises concerns. The measured I-Q distribution is the convolution of the signal and the added noise from the measurement circuit. The amplification chain begins with a HEMT, which introduces significant noise. The authors should clarify the impact of this noise on the results or present the I-Q distribution without the added noise.
5. How is the output power of the comb measured and calibrated? Could the authors provide more details on this aspect?
6. In Fig. 4c, the I-Q distributions for mode 11 in two different measurements do not overlap. Could the authors explain why this is the case?

7. From line 214 to line 220 & Fig. 4d, one of the main claims of this work pertains to the mutual coherence of the comb modes. Is it possible to perform a more detailed analysis of the mutual coherence of the modes? Specifically, in Fig. 4d, how well do the modes align to a line? A guide for the eye would be helpful. From the plot, it is not entirely clear that all the modes are perfectly aligned on the same line. Consequently, I am not convinced that the mutual coherence of the comb modes is well-established.

Reviewer #2

(Remarks to the Author)

I have reviewed the manuscript titled "Coherent microwave comb generation via the Josephson effect" that reports on a generation of equidistant, sharp, and coherent microwave tones at millikelvin temperatures using a magnetically periodically driven SQUID. There are a number of cryogenic microwave sources demonstrated in the past, including frequency combs, but the combination of magnetic drive and having the SQUID directly coupled to a broadband transmission line does not seem to be reported before for the purpose of microwave frequency combs of coherent tones. The paper is nicely written and is mostly not misleading. I list below a few points of criticism and provided that the authors can successfully revise their manuscript to address them, may be able to recommend the publication of this work in Nature Communications.

Points of criticism:

1) The abstract is lacking numeric key performance indicators of the frequency combs. Adding such will make it easier for researchers to compare the main results of this work to those of others. Lack of these will likely lead to incorrect citations to the work.

2) The abstract is advertising minimal dissipation of the device. However, no experimental data on the actual dissipation of the device is given anywhere in the manuscript. This is misleading. It should be relatively simple for the authors to estimate the power dissipated in millikelvin temperatures owing to their flux drive. I am referring to this dissipation since I feel that it will turn out to be the dominating source of dissipation, a source that is fully neglected in the analysis of the dissipation in the paper. It seems that the authors are driving the superconducting wire with milliampere-level of current which means a power of 100 microampere-level in a 50-ohm line. Although the authors do not use attenuators, the losses in the IR filters are often not smaller than 0.1 dB. This means that the power dissipated by their pump should be several microwatts, which is an astonishing 12 orders of magnitude higher than the attowatt-level number they quote in their manuscript for the internal dissipation of the SQUID. The authors should check these numbers, quote them in their manuscript for transparency and clarity, and possibly remove the claim about the low dissipation from their manuscript, especially from the abstract.

3) Related to the above comment, I still think that the authors do not need to necessarily remove completely their power dissipation analysis they have in their manuscript, but they need to make sure that it is just one source of dissipation that was not the main source of dissipation in the experiment. However, I have some points that could call for clarifications for their current analysis. Firstly, I do not see where the two microvolts for the voltage spike average height comes from. The authors say that they multiply the critical current with the normal-state resistance but if one does that, one obtains some hundreds of microvolts, not single microvolts. Secondly, I do not see where the 500 ps of average temporal width of the voltage spike comes from. It seems very long compared to the gigahertz pump frequency that the authors use in their experiment. Thirdly, I think that the 100 MHz pump frequency is not justified in this math since in reality, an order-of-magnitude higher pump frequency was used. Fourthly, why do the authors even do this kind of estimation using voltage spikes since they could do a rough estimation just based on the energy ($\sim 2E_J$) dissipated by the flux particle in an instantaneous flux sweep and the consequent phase flip of the SQUID, and they could just include a parallel resistor to their SQUID for their simulation (also attainable analytically) to get an accurate number for the fraction of energy dissipated in the SQUID internally. Well, I am not sure how accurate is the estimation of the subgap resistance of the authors since they have likely not measured it at least independently of having the 50-ohm shunt so I am not sure about the accuracy of this estimate, but it could be given provided that the assumptions and the relation to the experiment including the other channels of dissipation are clear.

3) In the introduction, the authors should also cite papers on dynamical Casimir effect [<https://www.nature.com/articles/nature10561>] and [<https://pubmed.ncbi.nlm.nih.gov/articles/PMC3600497>] since the devices used there are almost identical. Of course they are used in a different way but anyways such a discussion is in place. In addition, they could cite [<https://www.nature.com/articles/s41928-021-00680-z>] since it may be the millikelvin microwave source that provides the highest output power and also measures the phase noise spectrum and corresponding infidelity for quantum gates. The authors do not show the phase noise spectrum in their manuscript. Although such study would bring more value to this work, I do not see it absolutely necessary. However, the authors could cite this paper somewhere in the discussion part and say that in future experiments they could do such a study.

4) There are two samples, the parameters of which are spread out in the manuscript. It would be good to provide a table that has the parameters of the samples such as critical currents and also default parameters of the setup for each sample such as pump frequency and power. The authors could write that they use these default parameters if not otherwise explicitly stated. The authors should then check all figure legends and make sure that all the parameters are given for each panel. Currently, for example the pump power is not given for Fig. 2a although its frequency is given.

5) Also there is not enough information on the samples, namely, the various lengths the areas. If the authors do not have optical and SEM images, they could provide images of their design files and also add the design files to the data they

publish.

6) One of my most major concerns are the definitions of the microwave powers used in the manuscript. The authors often quote the pump and output powers of the device but they do not specify that at which point of the setup these power levels are defined. It seems to me that the pump power could be actually the power at the sample and if so that is good, except that it should be explicitly stated that it is so. For the output power, the case does not appear to be as clear. Based on my estimations, it may be that the authors quote the output power at room temperature after multiple amplifiers. If so, it is very misleading. The authors should to the best of their abilities provide the output powers of their different spectral lines at the output of the device so that their results can be compared with the other sources. The output power is one of the most important figures of merit of sources and lacking clarity on this is a major point of criticism.

7) Related to the above point, it also is rather misleading how the authors show their power spectral densities, i.e., they plot them in units of dBm, where power spectral density has units of dBm/Hz. The integral of the power spectral density in frequency over the peak provides then the total power in the tone.

8) Critical quantitative comparison of the obtained results with the previous literature is in place either already in the introduction, or perhaps better suited in the discussion. Namely, the linewidths, output powers, number of modes, dissipated power, power efficiency at millikelvin etc. should be compared with the previous results. Subsequently, the potential of this device and issues that still need to be solved before it will be useful in practical applications should be pointed out.

9) The authors seem to use the term Dynes parameter in a non-conventional way on line 430 and below it. Namely, they use it as is used in NIS junctions but they have an SIS junction where the parameter affects the subgap current in a different way (Dynes parameters of the left and right side should be multiplied together), but if the authors simply do not mention the term Dynes parameter, this point is fixed.

10) One of the Extended Data Fig. (maybe 2.) is missing a captions. In addition, the text is citing on line 166 to Extended Data Figs. 4-6 but I could find only four Extended Data Figs.

Minor points:

a) The first sentence in the introduction is incorrect with a trivial example of a sine wave of single frequency that it a periodic signal but only produces a single spectral line. In a frequency comb one wants to have a train of delta-function-like voltage pulses.

b) On line 30-32 it is claimed that in [14] the resonator restricts the frequency span of the harmonics, but that is not true. Consequently, finer report on the comparison of the obtained results to those of [14] is in place.

c) On line 88, it is said that no current flows through the SQUID but that is not true since there will be some current arising from the generated microwaves. This is just a wording issue. It could be said more clearly that that is an assumption for the following analysis.

d) On line 228, add word 'some' before 'other' since not in all previous works the statement applies.

e) On lines 234-237, it would be better to use mW:s instead of the powers of ten.

f) Remove 'down-' from line 295 since it is repeated later in the sentence.

g) 'conversion' should be spelled with s on line 305.

Reviewer #3

(Remarks to the Author)

This manuscript reports the generation of microwave frequency combs in a superconducting quantum interference device (SQUID) subject to a time-dependent magnetic drive. The periodic modulation produces a train of voltage pulses across the SQUID, which in the frequency domain corresponds to a comb spectrum with dozens of well-defined modes.

Low temperature frequency comb generation at microwave frequencies is an emerging field of interest for applications in cryogenic metrology, quantum information processing, and coherent microwave communications. The present work demonstrates a compact platform for coherent comb generation, with the comb's repetition rate easily tuned. The experiments are carefully executed, the device concept is straightforward, and the presentation is clear and concise. The manuscript should be of interest to the superconducting electronics, quantum technology, and precision measurement communities.

The manuscript presents a new method on a SQUID-based coherent microwave comb generator, supported by systematic measurements. I therefore support its publication in Nature Communications.

Below are a few comments for the authors to consider further to improve the clarity and completeness of this work.

1. Time-domain waveform: Since the comb arises from periodic voltage pulses across the SQUID, can the authors present

time-domain measurements? This would help directly relate pulse shape to the observed spectra within the detection bandwidth (or C-band frequency domain).

2. Repetition rate: The comb's repetition rate is determined by the drive frequency. The operations were demonstrated at two specific drive frequencies (1050 MHz and 390 MHz). How does the comb spectrum evolve with varying drive frequency? Is there a limit to the usable drive frequency?

3. Carrier-envelope offset frequency (f_{offset}): The f_{offset} is another critical parameter for the frequency comb. Could the authors determine the f_{offset} in their combs?

Version 2:

Reviewer comments:

Reviewer #1

(Remarks to the Author)

The authors have addressed most of my concerns and have made appropriate revisions to the manuscript. Given their detailed response and the accompanying revisions, I am inclined to recommend acceptance of this work for Nature Communications, pending clarification of one minor point: On line 174, should the condition read $\phi_{\text{ac}} > \phi_0$?"

Reviewer #2

(Remarks to the Author)

I have reviewed the revised manuscript titled "Coherent microwave comb generation via the Josephson effect", the reports of all reviewers, and the rebuttal of the authors. First, I would like to thank the authors for their thorough responses to the criticism and for their revisions to the manuscript. Most of the points of the reviewers seems to be now answered.

I have no strong scientific technical arguments any more, except that I would still give the power levels at the device since the room temperature values paly no significance for any possible utilization or analysis of the device. Thus they should be the ones hidden in the text, not the useful numbers even though they would bear more experimental uncertainty.

Again, the presentation language is nice and the experimental data seems clean. Good work on that. However, a more detailed list of the key performance indicators including efficiency and more objective comparison with other works could be in place. The authors seem to, for an unexplained reason, dislike injection locking, but they are themselves feeding in clocked pulses to their source at many orders of magnitude higher power than the injection locking tone used in previous works. It is well known that any source needs to be phase-locked or otherwise their phase will drift (against the frequency standard) sooner or later. To me it seems that the easier it is to deliver the clock, the more use cases such a device might find. Thus I needed to express some criticism here.

But the most critical point that I am pondering relates to the significance and impact of this work. Is it in (i) scientific novelty and innovation and/or (ii) technical improvement and potential practical impact? For (i), Reviewer 1 seemed critical saying that "This outcome, while technically sound, is not unexpected.". In my opinion, not all scientifically very important results need to be unexpected if they are the first demonstrations answering long-standing scientific questions. Is this the case here? I did not find convincing arguments to claim so. Thus I started to think that is this a major technical step towards new experiments and practical utility. The authors give an impression in the manuscript that this device could be useful in driving qubits, but in their response they admit that this is far-fetched. The other application they claim is qubit readout, which requires orders of magnitude less power than qubit driving. Ironically, the phase coherence is not necessarily so important in qubit readout. It is possible (although not experimentally demonstrated to my understanding) to read out qubits even with a fully incoherent radiation of narrow frequency band. In addition, even in qubit readout, the efficiency of the microwave source is important. The author say that they dissipate about one microwatt of power, i.e., -30 dBm, in their source at millikelvin, and yet they obtain from -170 dBm to -130 dBm of power out. That implies an extremely low efficiency or the order of 0.000000001, whereas previous works on microwave sources have already showed efficiencies of the order of 0.1. With such a low efficiency, it seems that the practical application of the source in scalable qubit readout is not immediate. Thus I am still not convinced about the point (ii) either.

In summary, I do not have major concerns on the scientific soundness of this work, but failed to find convincing arguments supporting its publication in terms of scientific of technological innovation and impact. In the end, this is a decision for the Editors of course.

Reviewer #3

(Remarks to the Author)

I have carefully examined the authors' responses to the reviewers' comments and the revised manuscript. The authors have addressed the main concerns raised in the initial round of review with substantial additional data, clearer explanations, and improved presentation. The revision significantly strengthens the manuscript, and in my opinion, it is now suitable for publication in Nature Communications.

Coherent microwave comb generation via the Josephson effect - Reply to Referees

We thank the Reviewers for their consideration and insightful comments to profitably improve the quality of the work. The criticisms have motivated us to perform a new run of measurements, which strengthen the results and further enlarge the functionalities demonstrated by our devices.

The point-by-point responses are detailed below and appear blue-colored. New or revised sentences in the manuscript file are red and appear red in the present reply file as well.

Figure 4 has been radically modified. Now it includes new I-Q plots to clarify the points raised by the Reviewers, and shows another important feature of the synthesized radiation: the phase tunability of the comb modes.

Additional new supporting material, including experimental data, simulations and pictures, appears in new sections in the Supplementary Information file.

With such important improvements, the revised manuscript reports a deeper analysis presented in a clearer way, and therefore we believe it warrants publication in Nature Communications.

REVIEWER 1

The manuscript by Angelo Greco et al. presents a novel approach to generating a microwave frequency comb based on superconducting circuits. The authors demonstrate that frequency combs with well-detuned comb spacing and a broad frequency range can be achieved by periodically modulating the flux across a SQUID. They characterize the comb emission by varying the magnetic flux modulation intensity and bias settings. The combs exhibit narrow linewidths and good coherence for the single mode, as well as well-defined mutual coherence among different modes, suggesting potential applications across a range of fields.

This work is grounded in the AC Josephson effect in SQUIDs, a well-established phenomenon that has been extensively reported in the literature. The authors employ a periodic signal to induce phase slips, which generate voltage pulses that naturally form a frequency comb. This outcome, while technically sound, is not unexpected. A significant portion of the manuscript focuses on characterizing the frequency comb in an attempt to demonstrate its excellent coherence. In principle, a train of ideal voltage pulses would yield an ideal frequency comb, which is also a well-known result. As such, the coherence of the experimentally synthesized frequency comb is primarily determined by the coherence of the flux drive signal and the stability of the DC bias. With this in mind, I have concerns regarding the characterization of the comb's coherence, as detailed in the following questions.

While the topic—generating microwave frequency combs with chip-scale devices- is certainly of broad interest, I am not convinced that the work presents a level of novelty that justifies publication in Nature Communications.

We thank the Reviewer for the comments on our manuscript and for acknowledging the relevance of the topic. In the following, we summarize the novelties of our comb generator which, we believe, justify the publication on Nature Communications. We have performed a comprehensive characterization of a device that:

- is cavity-free: the output has broad bandwidth, currently limited by setup components and cables
- has a small footprint: it is a single DC SQUID, with a few-micron squared loop area
- relies on standard Al-based SIS junctions by optical lithography, but the idea applies to combinations of other materials (for instance, Nb/Al-AlO_x/Nb) to allow higher temperatures operability
- generates highly coherent radiation, with a lower bound of few seconds, without employing any injection-locking technique
- never switches to the normal state, which makes it well-suited for direct (i.e., on-chip) integration with state-of-the-art quantum chips, with fragile systems susceptible to heat and quasi-particle poisoning.

On top of that, in the revised version we have demonstrated the phase tunability of the comb modes, which relies on the coherent nature of the generated radiation and represents a relevant asset for practical applications.

Additionally, I have several questions and points for further clarification:

1. Normalization of Phi Values: In the manuscript, the authors normalize ϕ_{DC} to ϕ_0 , but use units of dBm for ϕ_{AC} , which may cause confusion. Would it be possible to calibrate ϕ_{AC} also to ϕ_0 in order to provide a more consistent description?

We ourselves have been struggling deciding whether normalizing Φ_{ac} to Φ_0 or not; in the end we have concluded that it would be more correct to leave indicated the pump power. The reason for that is the following.

On the one hand, the calibration of Φ_{dc} is straightforward from the reflectometry data, where one can extract the mutual inductance between the flux line and loop, hence the periodicity of the SQUID characteristic and the normalized flux to Φ_0 .

On the other hand, calibrating Φ_{ac} can in principle be done from the two-dimensional plot in Figure 2c. From SPICE simulations we know that when Φ_{ac} induces a complete Φ_0 modulation an odd harmonic (like the 5th showed in Figure 2c) has no more its minimum at $\Phi_0/2$. As mentioned in the text, in Figure 2c we deduce that a Φ_0 modulation is achieved around -5 dBm. Nonetheless, in our opinion such an estimate is not accurate enough to rescale the y axis, since Device 2 does not allow to resolve the fine structures of the harmonic amplitude as a function of flux bias (differently from plot in Figure 2d for Device 1, whose flux line switches to the normal state before Φ_{ac} reaches Φ_0).

This concept is given in the text at lines 161-169, where the different Φ_{ac} regimes are explained together with the relative Φ_0 estimate.

2. is it $\phi_{AC} \gg \phi_0$?

This question finds an answer at line 167, where it is stated that -5 dBm is the threshold where $\Phi_{ac} = \Phi_0$. Above -5 dBm a time-dependent modulation of amplitude larger than one Φ_0 is induced.

3. Line 176: It would be helpful to list the value of ϕ_{AC} here as well, for completeness.

We have added the required value at line 190:

at a power of -17 dBm (meaning $\Phi_{ac} < \Phi_0$, as in Fig. 2b), with $\Phi_{dc} \simeq \Phi_0/2$.

4. From line 189 to line 193 & Fig. 3, the authors claim the coherence of a single mode of the frequency comb based on the shape of the I-Q histogram, which is one of the key results. However, this discussion raises concerns. The measured I-Q distribution is the convolution of the signal and the added noise from the measurement circuit. The amplification chain begins with a HEMT, which introduces significant noise. The authors should clarify the impact of this noise on the results or present the I-Q distribution without the added noise.

As argued by the Referee, the readout circuitry contributes to the recorded signal with additional photons generated by the cryogenic amplifier. Yet, a noise background calibration and detailed discussion on the emitted number of photons and their statistical properties are beyond the scope of this work. Nonetheless, the sizable width of the IQ distributions suggests the presence of broadening by thermal noise from the amplifier chain and other possible noise sources, as E_J fluctuations or pump instabilities.

A new section in Supplementary Information titled "Noise impact on IQ distributions" reports the discussion below and a new figure (Supplementary Fig. 6).

To appreciate the impact of the detection circuit, we compare the IQ distribution of a single mode when the drive tone is switched on and off. In the "pump off" condition, the generator emission spectrum is approximately zero, as having no resonator implies that there is neither cavity occupation due to thermal photons, nor a power spectral density emitted from a cavity. Therefore, the noise signal collected is mainly due the cold HEMT amplifier at the first stage of amplification.

Supplementary Figure 6a shows the IQ distribution of such signal ("pump off"), together with the data of the 9th harmonic of $f_p = 597$ MHz ("pump on"). Both histograms are obtained with the same demodulation parameters of Fig. 3c and Fig. 4b.

The two distributions are displaced from each other and are both nearly isotropic in the IQ plane. In Supplementary Figure 6b, one-dimensional cuts and Gaussian fits show that the two distributions have comparable widths, a hallmark of amplitude fluctuations at the generator level below our sensitivity.

5. How is the output power of the comb measured and calibrated? Could the authors provide more details on this aspect?

We thank for the question and recognize that a very similar criticism has been formulated by Referee 2 as well. We refer to Question 6 by Reviewer 2 below for the complete answer.

6. In Fig. 4c, the I-Q distributions for mode 11 in two different measurements do not overlap. Could the authors explain why this is the case?

As reported in the caption, the reason was that the 11th harmonic was sampled twice with two different LO tones, which caused an imperfect overlap of the spots at $I, Q \sim 0.2$ mV. However, the emended version of the manuscript presents a different plot (Fig. 4d), where the phase of 5 different modes has been addressed simultaneously and the dependence on the pump phase is discussed. We believe that this new set of measurements, together with Fig. 4b, provides new hints on the mathematical relation among the comb modes.

7. From line 214 to line 220 & Fig. 4d, one of the main claims of this work pertains to the mutual coherence of the comb modes. Is it possible to perform a more detailed analysis of the mutual coherence of the modes? Specifically, in Fig. 4d, how well do the modes align to a line? A guide for the eye would be helpful. From the plot, it is not entirely clear that all the modes are perfectly aligned on the same line. Consequently, I am not convinced that the mutual coherence of the comb modes is well-established.

We apologize if the analysis of the coherence of the comb modes was not clear enough. Triggered by the Referee's concern, we have acquired a new dataset.

Before discussing such new results, we would like to point out that the purpose of Figure 4 was to show how stable the phase relation among comb modes is over time by means of the dimensions of the spots of each mode in the I-Q plane. As the in-phase and quadrature signals come from the IQ mixer, the information of the actual phase of each mode was hidden by the quadrature sum $\sqrt{I^2 + Q^2}$ acquired after the second demodulation. This explains why all the spots lied on the same line in the graph of panel d in the old version.

As anticipated, we have addressed the Reviewer's comment by acquiring new data now presented in the revised version of Figure 4, panel b) and d). In the first two-dimensional histogram, panel b), we demonstrate the phase tunability of a single comb mode by means of the pump phase. Shortly, each mode posses a phase directly connected to the phase of the pump signal. Variations to the phase of the n -th mode, $\delta\theta_n$, can be imposed by changing the phase of the pump signal by $\delta\theta$ via a simple relation: $\delta\theta_n = n \times \delta\theta_p$. In the case of our two-step demodulation setup, the phase θ_p represents the phase of the pump signal with respect to the phase of the demodulators of the second down-conversion. In this sense, the stability of the phase relation between the modes primarily depends on the phase stability of the pump generator.

Panel d) relies on the feature just discussed, which we show to be valid across the whole spectrum by assessing the mathematical relation among the comb modes. Here the phases of 5 subsequent harmonics are addressed as a function of the pump phase, showing that they lay on straight lines with slope proportional to their mode number. By subtracting the phases of subsequent modes and fitting the data we verify that the relation $\delta\theta_n = n \times \delta\theta_p$ holds through all the analyzed spectrum, even at high modes orders.

In the emended version of the manuscript, a new paragraph titled "Phase control and mutual modes phase relation" presents the new Figure 4 and explains the concepts above.

In the following the reviewer can find the new paragraph added at lines 216 - 251:

We have demonstrated that the spectral modes of our frequency comb follow the simple relation of Eq. 2. It can be shown (see Methods) that in general a similar relation holds for the phase $\theta_n = n \times \theta_p + \tilde{\theta}_n$, where θ_n is the phase of the n -th mode, θ_p is the phase of the pump tone, and $\tilde{\theta}_n$ is an offset of the specific mode. This means that the phase of a mode varies according to the mode order when the pump phase is changed. We prove it by means of the setup shown in Fig. 4a, where a phase shifter to the pump tone θ_p is added. The generated comb signal is then acquired in both quadratures, similarly to Fig. 3b.

Figure 4b reports the IQ histograms of the 9th harmonic with $f_p = 597$ MHz for eight values of θ_p , shifted by 5 deg at each step. The eight resulting point clouds are rotated by 45 deg each other, implying that $\delta\theta_9 = 9 \times \delta\theta_p$ (δ denotes the phase difference).

This property holds for any harmonics of the comb spectrum. As examples, we choose a pump tone at $f_p = 121$ MHz, and we address the phase of the modes from 42 to 46 by the frequency-multiplexed demodulation setup in the schematic of Fig. 4c.

The experimental outcome is shown in Figure 4d, in terms of phase of such five harmonics as a function of the pump phase θ_p . The phase of each mode evolves linearly with respect to pump phase (to make it more visible, the phase data have been rescaled differently for each mode, while the intercepts $\tilde{\theta}$ have been shifted to zero).

A further consistency check is to verify that

$$\theta_{n+1} - \theta_n = \theta_p + \tilde{\theta}_{n+1} - \tilde{\theta}_n, \quad (1)$$

meaning that the phase difference between two subsequent modes is always equal to the pump phase θ_p plus an offset. The inset of Fig. 4d shows $(\theta_{43} - \theta_{42})$ vs θ_p (offset removed for clarity). The linear fit yields a slope of 1, disclosing the constraint between the phases of the harmonics (Eq. 3), which, together with the equidistance of frequencies (Eq. 2), is peculiar of a comb spectrum.

REVIEWER 2

I have reviewed the manuscript titled "Coherent microwave comb generation via the Josephson effect" that reports on a generation of equidistant, sharp, and coherent microwave tones at millikelvin temperatures using a magnetically periodically driven SQUID. There are a number of cryogenic microwave sources demonstrated in the past, including frequency combs, but the combination of magnetic drive and having the SQUID directly coupled to a broadband transmission line does not seem to be reported before for the purpose of microwave frequency combs of coherent tones. The paper is nicely written and is mostly not misleading. I list below a few points of criticism and provided that the authors can successfully revise their manuscript to address them, may be able to recommend the publication of this work in Nature Communications.

Points of criticism:

1. The abstract is lacking numeric key performance indicators of the frequency combs. Adding such will make it easier for researchers to compare the main results of this work to those of others. Lack of these will likely lead to incorrect citations to the work.

We thank the Reviewer for the comment. We have added some parameters of our frequency comb which will help the readers in highlighting key aspects, like emitted power at device output, dynamic range, number of harmonics, and footprint. In the following the new part in the modified abstract:

A time-dependent magnetic drive periodically generates voltage pulses, which in the frequency domain correspond to a comb with dozens of spectral modes here reported up to mode 46. The emitted power at the device level ranges from -170 dBm to -130 dBm per harmonic, corresponding to 40 dB dynamic range in the 4-8 GHz bandwidth.

2. The abstract is advertising minimal dissipation of the device. However, no experimental data on the actual dissipation of the device is given anywhere in the manuscript. This is misleading. It should be relatively simple for the authors to estimate the power dissipated in millikelvin temperatures owing to their flux drive. I am referring to this dissipation since I feel that it will turn out to be the dominating source of dissipation, a source that is fully neglected in the analysis of the dissipation in the paper. It seems that the authors are driving the superconducting wire with milliampere-level of current which means a power of 100 microampere-level in a 50-ohm line. Although the authors do not use attenuators, the losses in the IR filters are often not smaller than 0.1 dB. This means that the power dissipated by their pump should be several microwatts, which is an astonishing 12 orders of magnitude higher than the attowatt-level number they quote in their manuscript for the internal dissipation of the SQUID. The authors should check these numbers, quote them in their manuscript for transparency and clarity, and possibly remove the claim about the low dissipation from their manuscript, especially from the abstract.

We thank the Reviewer for pointing out this aspect which appeared misleading. In the previous version of the paper, we have extensively discussed the dissipation due to Joule heating by the two Josephson junctions per pulse, in analogy with analyses on other pulse emitters (e.g., Mukhanov, IEEE Transaction on Applied Superconductivity, Vol. 21, No. 3 (2011)). Such figure of merit is of primary relevance to interface on-chip the comb source with other quantum systems, such as qubits or sensors, sensitive to quasi-particles generation. However, as argued by the Reviewer, additional sources of dissipation are present when considering the setup beyond the SQUID itself, such as the static power dissipation associated to the current across the resistive network in the drive line circuitry. As the Referee points out, the dc flux bias is provided to the sample by a coaxial line with LC and IR filters interposed at the mixing chamber stage. The ohmic resistance of the flux line is 40Ω from room temperature to the mixing chamber plate, including semi-attenuated coax cables, low-pass filters and IR filters. With a current bias of $100 \mu\text{A}$, this yields a dissipated power of $0.4 \mu\text{W}$ across the whole

cryostat.

The ac component of the flux is added to the dc part out of the fridge, so that the same line conveys both dc and ac signal to the sample, see the setup schematic. The coaxial cables from room temperature down to the coldest plate attenuate 14-20 dB within the 0.2-1 GHz bandwidth. At the mixing chamber stage, after low-pass and infrared filtering, copper coaxial lines connect the chip carrier with negligible attenuation. Altogether, in this case the dissipation is of the order of tens of μW throughout the fridge, with much less than $1\mu\text{W}$ at the mixing chamber plate (a 0.1 dB dissipation by a IR filter of a -20 dBm signal yields a dissipation of $0.23\mu\text{W}$). It is worth mentioning that a way to reduce the current through the drive line, and therefore the overall dissipation, is to enhance the magnetic coupling of the flux line to the SQUID loop, for instance by on-chip spiral inductors patterned on top of the SQUID (Chen et al., Journal of Semiconductor Technology and Science 4, 149 (2004)).

We have amended the second last sentence and extended the Methods section to summarize the considerations above.

Main file, lines 264-280: The micrometer-size and the number of generated modes make our platform suitable for scalability, while the minimal dissipated power allows on-chip integration with other quantum systems sensitive to quasi-particles generation, such as qubits or sensors. In the present experiment, the power dissipation at the coldest plate is primarily due to the flux drive ($1\mu\text{W}$ at most, see Methods), which can be reduced by enhancing the inductive coupling between flux line and SQUID loop, while the contribution of the device is negligible. Such a heat load is largely sustainable by current dilution refrigerators with typical cooling power ranging from from 0.1 mW to few mW (Pauka et al, Nature Electronics 4, 64 (2021)).

Methods: The dissipation due to Joule heating by the two Josephson junctions per pulse is a figure of merit of primary relevance to interface on-chip the comb source with other quantum systems. However, additional sources of dissipation are present when considering the setup beyond the SQUID itself, such as the static power dissipation associated to the current across the resistive network in the drive line circuitry. For instance, the dc flux bias is provided to the sample by a coaxial line with LC and IR filters interposed at the mixing chamber stage, see Supplementary Figure XXX. The ohmic resistance of the flux line is 40Ω from room temperature to the mixing chamber plate, including semi-attenuated coax cables, low-pass filters and IR filters. A typical current bias of $100\mu\text{A}$ yields a dissipated power of $0.4\mu\text{W}$ across the whole cryostat.

About the ac component of the flux, the coaxial cables from room temperature down to the coldest plate attenuate 14-20 dB within the 0.2-1 GHz bandwidth. At the mixing chamber stage, after low-pass and infrared filtering, copper coaxial lines connect the chip carrier with negligible attenuation. Altogether, in this case the dissipation is of the order of tens of μW throughout the fridge, with much less than $1\mu\text{W}$ at the mixing chamber plate (a 0.1 dB dissipation by a IR filter of a -20 dBm signal yields a dissipation of $0.23\mu\text{W}$).

It's worth mentioning that a way to reduce the current through the drive line, and therefore the overall dissipation, is to enhance the magnetic coupling of the flux line to the SQUID loop, for instance by on-chip spiral inductors patterned on top of the SQUID (Chen et al., Journal of Semiconductor Technology and Science 4, 149 (2004)).

- 3a. Related to the above comment, I still think that the authors do not need to necessarily remove completely their power dissipation analysis they have in their manuscript, but they need to make sure that it is just one source of dissipation that was not the main source of dissipation in the experiment. However, I have some points that could call for clarifications for their current analysis. Firstly, I do not see where the two microvolts for the voltage spike average height comes from. The authors say that they multiply the critical current with the normal-state resistance but if one does that, one obtained some hundreds of microvolts, not single microvolts. Secondly, I do not see where the 500 ps of average temporal width of the voltage spike comes from. It seems very long compared to the gigahertz pump frequency that the authors use in their experiment. Thirdly, I think that the 100 MHz pump frequency is not justified in this math since in reality, an order-of-magnitude higher pump frequency was used. Fourthly, why do the authors even do this kind of estimation using voltage spikes since they could do a rough estimation just based on the energy ($\sim 2E_J$) dissipated by the flux particle in an instantaneous flux sweep and the consequent phase flip of the SQUID, and they could just include a parallel resistor to their SQUID for their simulation (also attainable analytically) to get an accurate number for the fraction of energy dissipated in the SQUID internally. Well, I am not sure how accurate is the estimation of the subgap resistance of the authors since they have likely not measured it at least independently of having the 50-ohm shunt so I am not sure about the accuracy of this estimate, but it could be given provided that the

assumptions and the relation to the experiment including the other channels of dissipation are clear.

We agree with the Reviewer, the Joule heating at the device level is definitely not the main source of the dissipation in the experiment, as discussed in the point above.

About the first concern aforementioned, the height of the voltage pulses is inferred from the output power at the device level (see Question 6 below) and qualitatively supported by JoSIM simulations. Supplementary Note 3 now contains a subsection where the predicted voltage pulses are reported, including their Fourier transform yielding the comb spectrum.

About the second concern, the estimate of the temporal width of the voltage pulses is a complicated subject experimentally. Previous detailed calculations in Ref. [21] (Bosisio, et al., Journal of Applied Physics 118, 213904 (2015)) have shown that it depends on the intrinsic properties of the junctions, such as superconducting material, loop geometrical inductance, junctions normal state resistance, capacitance and asymmetries. In particular, Ref. [21] predicts voltage pulses with few tens of ps width with a 1 GHz flux drive. However, it's shown that SQUIDs with asymmetric junctions and sizable capacitance, as in our experiment, lead to broader pulses than the ideal case. On top, in realistic implementations, the environment the device is embedded in, such as rf lines and circuitry with limited bandwidth, damps and smears the shape of the voltage pulses. Again, numerical calculations by JoSIM account for the junctions parameters predict voltage pulses with few-hundred picoseconds duration. The plots are shown in Supplementary Note 3.

Third concern: 100 MHz is the order of magnitude of the pump tone frequency for most of the dataset reported. f_p above 1 GHz has been used only for panel a in Figure 2. According to simulations, both height and width of the pulses vary with the pump frequency, though remaining in the few μV and hundreds of ps range, respectively. Please see again Supplementary Note 3.

Forth concern. It should be clarified that in the washboard potential description the flux modulation does not cause the phase particle to overcome the $2E_J$ energy barrier separating adjacent minima. Instead, such barrier is lowered by the time-dependent flux, and the phase particle "rolls down" toward the minimal energy according to its dynamics (we prefer not to refer to these periodic phase variations as phase slips, where instantaneous phase jumps quantized in units of 2π are accompanied by the formation and reabsorption of a gapless center, neither of which happen in our case). In the following, we qualitatively describe the dynamics of a SQUID with moderate asymmetry over the first semi-period of $\Phi_{ac}(t)$ by a washboard model. As the flux modulation begins, the phase particle is lifted up in energy while slightly advancing in the phase (φ) axis till $\Phi(= \Phi_{dc} + \Phi_{ac}) < \Phi_0/2$; so far, $d\varphi/dt$ is minimal (zero in the case of symmetric SQUID), and so the voltage across the SQUID and hence the dissipation. As a result, the electromagnetic energy density has increased by E_J and has caused circulating (dissipationless) currents in the SQUID loop. When $\Phi \approx \Phi_0/2$, the phase particle rolls leading to a dissipation $\simeq V^2/R_{sg}$ and the circulating currents are inverted. Finally, when $\Phi > \Phi_0/2$, the particle is pulled down in the new energy minimum, again with a small phase advance. The first semi-period is now concluded, and the cycle starts again. The sign of the ac flux and the direction of the phase rolling will be inverted.

Last concern. The estimation of the subgap resistance R_{SG} comes from DC characterizations of test devices realized using the same fabrication technique used for the present work. These measurements highlighted values for the subgap resistance of $\frac{R_{SG}}{R_N} = 10^2 - 10^3$, which we use to estimate the dissipated power on the device. These values are coherent with the ones reported in literature, where also values of $\frac{R_{SG}}{R_N} = 10^4$ can be found (M. A. Gubrud et al., IEEE Transactions on Applied Superconductivity, vol. 11, no. 1, pp. 1002-1005, (2001)).

- 3b. In the introduction, the authors should also cite papers on dynamical Casimir effect [https://www.nature.com/articles/nature10561?utm_source=chatgpt.com] and [https://pmc.ncbi.nlm.nih.gov/articles/PMC3600497/?utm_source=chatgpt.com] since the devices used there are almost identical. Of course they are used in a different way but anyways such a discussion is in place. In addition, they could cite [<https://www.nature.com/articles/s41928-021-00680-z>] since it may be the millikelvin microwave source that provides the highest output power and also measures the phase noise spectrum and corresponding infidelity for quantum gates. The authors do not show the phase noise spectrum in their manuscript. Although such study would bring more value to this work, I do not see it absolutely necessary. However, the authors could cite this paper somewhere in the discussion part and say that in future experiments they could do such a study.

We thank the Reviewer for suggesting valuable references. We have inserted these works and two additional citations in the introduction at line 16.

- There are two samples, the parameters of which are spread out in the manuscript. It would be good to provide a table that has the parameters of the samples such as critical currents and also default parameters of the setup for each sample such as pump frequency and power. The authors could write that they use these default parameters if not otherwise explicitly stated. The authors should then check all figure legends and make sure that all the parameters are given for each panel. Currently, for example the pump power is not given for Fig. 2a although its frequency is given.

We thank the Reviewer for the useful comment. To this end, we add in the Methods, under the Device Fabrication section, a recap of the main features of both devices, in order to give the reader a clearer view of the presented devices. Here the related text:

By construction in both devices, the self-inductance of the SQUID loop is 100 pH, while the critical currents range between 100 nA and 200 nA (range given by the fabrication dispersion of tunnel junctions parameters.) Sample 1 has a mutual inductance between the flux line and the loop of 4.5 pH, while sample B of 0.44 pH. The difference in mutual inductance is obtained by modifying the width of the flux line next to the SQUID loop. We have corrected the captions in the text adding the pump parameters when needed.

- Also there is not enough information on the samples, namely, the various lengths the areas. If the authors do not have optical and SEM images, they could provide images of their design files and also add the design files to the data they publish.

We thank the Referee for pointing this lack out. We are glad to add the CAD of a typical chip containing two identical devices, together with a SEM image of the SQUID loop plus flux line in the supplementary information file.

- One of my most major concerns are the definitions of the microwave powers used in the manuscript. The authors often quote the pump and output powers of the device but they do not specify that at which point of the setup these power levels are defined. It seems to me that the pump power could be actually the power at the sample and if so that is good, except that it should be explicitly stated that it is so. For the output power, the case does not appear to be as clear. Based on my estimations, it may be that the authors quote the output power at room temperature after multiple amplifiers. If so, it is very misleading. The authors should to the best of their abilities provide the output powers of their different spectral lines at the output of the device so that their results can be compared with the other sources. The output power is one of the most important figures of merit of sources and lacking clarity on this is a major point of criticism.

We apologies if pump power and output power have not been defined clearly. We have referred to the pump power as the "nominal power", i.e. the power at rf generator output prior to room temperature and cryogenic attenuation. As shown in the setup schematic (Supplementary Figure 5), such a signal is firstly attenuated at room temperature by -15 dB, and secondly by coaxial cables of the cryostat from outer connectors down to the mixing chamber plate (attenuation from -14 dB to -20 dB which scales linearly in frequency in the 0.2-1 GHz bandwidth). These numbers now appear clearly in the Methods (see the answer to Question 2 above). We have also added a reference in the text to the Methods section to underline the concept.

About the emitted power, the data reported refer to the power acquired at room temperature, and not the sample output. We made this choice because at the device output the power can be just estimated, while the experimental data we have access to is at the acquisition instrument. Nonetheless, we agree that the knowledge of the power at the sample level would be a better benchmark of the emission of our source. A new sentence at line 136 clearly indicates how to rescale down the numbers in the graphs to get a good approximation of the emitted power at the device level:

Later on, we shall refer to the harmonics power as the power recorded at the acquisition step, i.e. the power emitted by the device plus an instrumental gain which varies approximately linearly from ~ 87 dB at 4 GHz to ~ 82 dB at 8 GHz. See Methods and Supplementary Note 4 for further details on the setup and the amplification path.

Finally, in Supplementary Note 4 we detail the amplification steps from the sample to the acquisition:

To properly rescale the measured power shown in the plots to the power emitted to the sample output, we have to quantify the amplification along the readout line. As argued in the main text, the overall gain of the output line from output port of the sample to instrumental acquisition ranges between 87 dB at 4 GHz and 82 dB at 8 GHz (indicated for brevity as 87-82 dB). Such numbers include 30-20 dB amplification in the same 4-8 GHz

bandwidth across the output line of the cryostat (copper coax from sample to circulator, TiN superconducting wire from circulator to HEMT, HEMT, and CuNi coax from HEMT to output connector) and the further two-stage room temperature amplification (28 dB - 3 dB + 33 dB).

7. Related to the above point, it also is rather misleading how the authors show their power spectral densities, i.e., they plot them in units of dBm, where power spectral density has units of dBm/Hz. The integral of the power spectral density in frequency over the peak provides then the total power in the tone.

We thank the Reviewer for the question. Through the entire paper the shown spectra have on the y axis the unit dBm rather than dBm/Hz because what we plot is actually the power spectrum (PS) rather than the power spectral density (PSD). Our harmonics have indeed a very narrow linewidth, which is always lower than the resolution bandwidth (RBW) of our spectrum analyzer. This means that, regardless the RBW we use, the comb lines always look like deltas and always have the very same amplitude, since all the power is concentrated into a narrow frequency bin.

The PSD, which has the same physical information of the PS, would show peaks with an amplitude dependent on the normalization (the RBW).

We add that typically the PSD is used for spectra where the signal is distributed on a wide portion of the spectrum, while the PS can be effectively used when dealing with delta-like signals, just like in our case.

8. Critical quantitative comparison of the obtained results with the previous literature is in place either already in the introduction, or perhaps better suited in the discussion. Namely, the linewidths, output powers, number of modes, dissipated power, power efficiency at millikelvin etc. should be compared with the previous results. Subsequently, the potential of this device and issues that still need to be solved before it will be useful in practical applications should be pointed out.

The intrinsic narrow linewidth of our spectral modes aligns with the one observed for other states that do not couple to a dissipative reservoir (Erickson et al., PRL 113, 187002 (2014)). However, our architecture relies on a different working principle and generates a comb spectrum without any resonator to be seeded to accomplish four-wave mixing (Erickson et al., PRL 113, 187002 (2014)) or whose emission has to be stabilized via injection-locking (Cassidy et al., Science 355, 939 (2017), and Wang et al., Nature Communications 15, 2041 (2024)).

The output power at device level is estimated between -170 dBm and -130 dBm in the 4-8 GHz band. Such a power could easily inject few photons into microwave resonators placed along the output line, thus realizing a cryogenic-integrated tool for qubit readout with good fidelity or quantum sensing. The possibility to drive coherent quantum operations looks more challenging for superconducting microwave sources at a general level. Preliminary demonstrations of qubit coherent control have been presented (Bao, et al., Nature Communications 15, 5958 (2024)), though for this purpose superconducting emitters still provide limited frequency spans on the generated radiation and need further integration with other components (as discussed in Yan et al., Nature Electronics 4, 885 (2021)). On this last regard, our device represents an extremely flexible solution as it broadens the emitted power over a few GHz bandwidth with an adjustable pump tone. So, even though the qubit drive looks challenging to implement due to the frequency and phase constraints between the modes of a comb spectrum, at the same time such features may provide other functionalities in the field of quantum information processing (Hayes et al., PRL 104, 140501 (2010)). Strategies to boost the emitted power include the employment of SQUIDS with tunable symmetry, for instance via electrostatic gating of semiconducting weak links, the use of other geometries as rf SQUIDS, where the phase-flux relation around half flux quantum can be very steep next to the crossover to the hysteretic regime, and via flux modulation by squared pulses.

We have summarized such considerations at from line 297:

Contrary to previous demonstrations, the comb spectrum is generated without any resonator to be seeded to accomplish four-wave mixing (Erickson et al., PRL 113, 187002 (2014)) or whose emission has to be stabilized via injection-locking (Cassidy et al., Science 355, 939 (2017), and Wang et al., Nature Communications 15, 2041 (2024)). The tunability of the comb emitted opens up the possibility of exploiting optical techniques, e. g. frequency comb spectroscopy (Picque et al., Nature Photonics 13, 146 (2019)), to selectively address integrated sensors, and perform multiqubit entangling operations (Hayes et al., PRL 104, 140501 (2010)) or readout. In the future, we aim to boost the output power by using dc SQUIDS with tunable symmetry (recall Fig. 1b), for instance via electrostatic gating of semiconducting weak links, or by means of other geometries as rf SQUIDS, where the phase-flux relation around half flux quantum can be very steep next to the crossover to the hysteretic

regime, and finally via optimized flux signals.

9. The authors seem to use the term Dynes parameter in a non-conventional way on line 430 and below it. Namely, they use it as is used in NIS junctions but they have an SIS junction where the parameter affects the subgap current in a different way (Dynes parameters of the left and right side should be multiplied together), but if the authors simply do not mention the term Dynes parameter, this point is fixed.

We thank the Reviewer for this correction.

The previous definition of subgap resistance is in fact incorrect referred to the Dynes parameter. Since this aspect is not crucial for the paper, as suggested, we just do not mention it.

10. One of the Extended Data Fig. (maybe 2.) is missing a captions. In addition, the text is citing on line 166 to Extended Data Figs. 4-6 but I could find only four Extended Data Figs.

We apologies for the misprint, there were indeed only four Extended Figures in the previous version of the work. The current Supplementary File contains the old Extended Data plots together with new figures and new paragraphs.

Minor points:

- a) The first sentence in the introduction is incorrect with a trivial example of a since wave of single frequency that it a periodic signal but only produces a single spectral line. In a frequency comb one wants to have a train of delta-function-like voltage pulses.
The Referee is right, this is a misprint. We have rephrased the starting sentence as follows (lines 4-6): **Time-frequency duality implies that a periodic signal in time corresponds to one or more equally spaced spectral lines in the frequency domain.**
- b) On line 30-32 it is claimed that in [14] the resonator restricts the frequency span of the harmonics, but that is not true. Consequently, finer report on the comparison of the obtained results to those of [14] is in place.
We agree with the Reviewer. In Wang et al., Nat. Com. 15, 2041 (2024), the harmonic frequencies remain bounded to multiples of the resonator frequency, which is not tunable; however, the harmonics can span higher frequencies indeed. We have modified the sentence as follows:
... and constrict the frequency of the harmonics.
- c) On line 88, it is said that no current flows through the SQUID but that is not true since there will be some current arising from the generated microwaves. This is just a wording issue. It could be said more clearly that that is an assumption for the following analysis.
Thank you for the careful reading. Now the sentence reads: **.. since no dc bias current is imposed across the SQUID..**
- d) On line 228, add word 'some' before 'other' since not in all previous works the statement applies.
Thank you, done.
- e) On lines 234-237, it would be better to use mW:s instead of the powers of ten.
If the Referee meant to express the numbers in "mW per second" units, we have changed the sentence about the cryo-CMOS circuits as follows:
This offers a significant advantage over cryogenic complementary metal oxide semiconductor (cryo-CMOS) circuits, whose dissipation is of the order of tens of mW for 1 GHz clock frequencies.
- f) Remove 'down-' from line 295 since it is repeated later in the sentence.
We agree, done.
- g) 'conversion' should be spelled with s on line 305.
Thank you, the typo has been fixed.

REVIEWER 3

This manuscript reports the generation of microwave frequency combs in a superconducting quantum interference device (SQUID) subject to a time-dependent magnetic drive. The periodic modulation produces a train of voltage

pulses across the SQUID, which in the frequency domain corresponds to a comb spectrum with dozens of well-defined modes.

Low temperature frequency comb generation at microwave frequencies is an emerging field of interest for applications in cryogenic metrology, quantum information processing, and coherent microwave communications. The present work demonstrates a compact platform for coherent comb generation, with the comb's repetition rate easily tuned. The experiments are carefully executed, the device concept is straightforward, and the presentation is clear and concise. The manuscript should be of interest to the superconducting electronics, quantum technology, and precision measurement communities.

The manuscript presents a new method on a SQUID-based coherent microwave comb generator, supported by systematic measurements. I therefore support its publication in Nature Communications.

Below are a few comments for the authors to consider further to improve the clarity and completeness of this work.

1. Time-domain waveform: Since the comb arises from periodic voltage pulses across the SQUID, can the authors present time-domain measurements? This would help directly relate pulse shape to the observed spectra within the detection bandwidth (or C-band frequency domain).

We made quite some attempts to record time-domain signals prior to the demodulation to baseband or acquisition via the power spectrum analyzer, even with narrow pass-band filtering. Unfortunately, such time traces have not allowed to appreciate the shape of the emitted voltage pulses due to noisy contributions distributed across the acquisition bandwidth. On the contrary, the comb spectrum clearly stands out in the frequency trace, as appears from the plots in Figure 2.

2. Repetition rate: The comb's repetition rate is determined by the drive frequency. The operations were demonstrated at two specific drive frequencies (1050 MHz and 390 MHz). How does the comb spectrum evolve with varying drive frequency? Is there a limit to the usable drive frequency?

We thank the Reviewer for underlining this important point. The two spectra shown in Fig. 2 are only representative of two typical frequency pump regimes, namely the 100 MHz and 1 GHz orders of magnitude. The flux line used to pump the SQUID is low-pass filtered at about 1.5 GHz, meaning that in the current setup we can not use higher frequencies.

By raising the pump frequency the spectrum changes mainly in two ways. First, the spectral lines change their position and spacing accordingly with the comb equation $f_n = n \cdot f_p$, where f_n is the frequency of the n-th and f_p is the pump frequency. Second, by increasing the pump frequency the generated voltage spikes are sharper and shorter since $V \propto \frac{d\varphi}{dt}$. This means that the power is spread differently through the whole spectrum, in particular at higher frequencies.

There are some limitations concerning both the maximum harmonic frequency and pump frequency usable for the ac flux. A first limit concerns the depairing frequency, i.e. the photon frequency at which the Cooper pairs are broken, an intrinsic limitation by the materials. For Aluminum it is about 50 GHz, while for Niobium can be higher than 300 GHz; as such, a wise choice of the superconducting material can raise the frequency of the generated harmonics. A second limit is given by the circuit parameters of the SQUID, and concerns the highest usable pump frequency. A precise treatment is not trivial, as it involves the inductance of the loop, junction capacitance, critical current and shunt resistance, together with the electromagnetic environment. To provide an estimation of the intrinsic limits restricted to the physics of the SQUID itself, we can consider its plasma frequency. This bottleneck regards the capability of the phase particle (intended in a washboard potential picture) to follow rapid flux variations in the loop. In other words, the flux in the SQUID can be tuned up to timescales set by the plasma frequency (Kokkonen et al., Scientific Reports 7, 14713 (2017)), which can be routinely set to be over 100 GHz (Lotkhov et al., Appl. Phys. Lett. 115, 192601 (2019)). In experimental works about flux-pumped parametric amplification (Grebel et al., Appl. Phys. Lett. 118, 142601 (2021)) and photon emission (Wilson et al, Nature, 479, 376–379 (2011)), SQUIDs frequencies of tenth of GHz are commonly used. We therefore expect to be able to operate up to those frequencies at least.

We have added a comment regarding these frequency limits from line 281:

There are then some limitations concerning both the maximum harmonic frequency and pump frequency usable for the ac flux.

The former is limited by the depairing frequency, i.e. the photon frequency at which the Cooper pairs are broken, of the superconductor used to realize the device. It is directly related to the superconducting gap, and as an example can be about 50 GHz for aluminum and higher than 300 GHz by using niobium technology.

The latter, is mainly related to the plasma frequency of the SQUID. This bottleneck regards the capability of the SQUID phase to follow rapid flux variations in the loop, which can be tuned up to timescales set by the plasma frequency. The plasma frequency of SQUIDs routinely used can be set between 10 GHz to over 100 GHz.

3. Carrier–envelope offset frequency (f_{offset}): The f_{offset} is another critical parameter for the frequency comb. Could the authors determine the f_{offset} in their combs?

We thank the Reviewer for the question, which allows us to highlight an important difference between our comb and conventional optical combs. In combs generated by mode-locked laser systems, for instance, the train of pulses is emitted at the output of a cavity. The cavity dimension and group velocity of the light set the carrier frequency and the pulses repetition rate, while the dispersion inside the cavity determines the phase shift between the carrier and the envelope. The latter, in particular, leads to a finite carrier–envelope offset in the frequency domain. In our case, the carrier–envelope offset frequency is zero by construction as there’s no cavity and, therefore, no carrier frequency inside the voltage pulses to which refer a phase offset.

We have clarified this point as follows (line 46):

We emphasize that pulse synthesis here does not rely on cavities, unlike conventional optical frequency combs. This leads to few key differences. In cavity-based systems, cavity length and group velocity set the carrier frequency and the pulses repetition rate, whereas in our case the pump frequency f_p is freely tunable, which potentially allows harmonic generation from gigahertz to terahertz. Moreover, while cavity dispersion induces a finite carrier–envelope frequency offset in the spectrum, our architecture yields an offset frequency of zero, since there’s no cavity and, therefore, no carrier frequency inside the voltage pulses.

We would like to thank once more the Editor and the Reviewers for recognizing the improved quality of the manuscript and endorsing the publication on Nature Communications.

About the concern raised by Referee 1 in the latest round of correspondence, we clarify that the condition $\Phi_{ac} \ll \Phi_0$ at line 174 is correct. It indeed refers to the plot in Fig. 2d, which is a magnification of the signal structure around half flux quantum for small drive amplitudes, i.e. $\Phi_{ac} \ll \Phi_0$. In this regime, the one-dimensional traces show the interference pattern of Fig. 2e, which are in excellent agreement with the predictions by our simulations (fig. 2f), as discussed in the text. We finally point out that Sample 2 was conceived for such high resolution scans on Φ_{dc} , as designed with a loop-flux line inductive coupling one order of magnitude smaller than in Sample 1. In fact, on Sample 2 we could not reach the regime $\Phi_{ac} \gg \Phi_0$ as the flux line was switching to the normal state (and so the SQUID closeby) at $\Phi_{ac} \sim \Phi_0$.